# Efficacy of ceiling-mounted mosquito nets for malaria vector control in a Peruvian Amazon riverine community: A stepped-wedge cluster randomized trial

Antonio Marty Quispe[1,2*], Carlos Álvarez-Antonio[3], Freddy Gutierrez Rodriguez[3], Cristiam Carey[4], Hugo Rodríguez-Ferrucci[5], Cinthia Inti[6], Eun Seok Kim[6,7]

**1** Escuela de Posgrado, Universidad Señor de Sipán, Chiclayo, Peru, **2** BIO, Universidad de Ingeniería y Tecnología, Lima, Peru, **3** Gerencia Regional de Salud de Loreto, DIRESA, Loreto, Peru, **4** Universidad Nacional de la Amazonía Peruana, Loreto, Peru, **5** World Vision Peru, Loreto, Peru, **6** World Vision Korea, Seoul, Korea, **7** London School of Hygiene and Tropical Medicine, London, United Kingdom

* drantonioquispe@gmail.com

## Abstract

Malaria remains a major global health concern, especially in tropical regions such as the Peruvian Amazon. Installing ceiling-mounted mosquito nets in houses has been proposed as a strategy to reduce mosquito–human contact and, in turn, lower malaria transmission. We aimed to assess the effectiveness of ceiling-mounted mosquito nets in reducing Anopheles mosquito density in a high-transmission Amazonian setting. We conducted a stepped-wedge cluster-randomized trial from March to December 2024 in the Llanchama community of the San Juan Bautista district, Loreto, Peru. A total of 69 households were randomized into three clusters, receiving the intervention at staggered three-month intervals. We measured entomological indices using human landing catches (HLC) and other standardized methods, focusing on Anopheles mosquito counts, bites per person per night (BPN), and bites per person per hour (BPH), both indoors and peridomestic, derived from the same HLC sessions, using cumulative mosquito captures over 12-hour periods to compute biting indices. Houses were retrofitted with mosquito nets across ceilings and other open structural areas, creating a barrier to prevent mosquito entry and lower exposure. Analysis followed a stepped-wedge mixed-effects negative binomial model adjusting for clustering and time trends. Our per-protocol analysis shows that, compared to non-renovated households, the renovated households experienced a 55% reduction in indoor Anopheles counts (95% CI: 33%–74%; p=0.004), a 60% decrease in indoor BPN (95% CI: 27%–78%; p=0.003), and a 61% reduction in indoor BPH (95% CI: 15%–83%; p=0.018). Peridomestic mosquito counts, BPN, and BPH did not differ significantly between renovated and non-renovated households. Our study provides evidence that installing mosquito nets across household ceilings markedly reduces indoor Anopheles presence and biting rates, suggesting that this structural

**Data availability statement:** The anonymized, aggregated data collected for this study is available under the Creative Commons Attribution 4.0 (CC BY 4.0) license and freely available with the following doi: https://dx.doi.org/10.6084/m9.figshare.28830449.

**Funding:** This study was conducted under the Reduction of the Burden of Neglected Tropical Diseases and Promoting Basic Health in Peruvian Amazonia (RTA) project, which was financially supported by the Korea International Cooperation Agency (KOICA). Publication of this article involved collaboration with KOICA, World Vision (Grant No. 2021-067), and the Regional Health Management of Loreto. World Vision Korea received the funds from KOICA (https://www.koica.go.kr/sites/koica_en/index.do). The funders had no role in the study design, data collection and analysis, decision to publish, or preparation of the manuscript.

**Competing interests:** The authors declare no conflict of interest.

modification could be a promising strategy for lowering malaria risk in riverine communities.

## Introduction

Malaria remains a major global health concern, disproportionately affecting tropical regions like the Peruvian Amazon, where environmental conditions favor the proliferation of Anopheles mosquitoes, the primary vectors of the disease [1]. Traditional measures, such as insecticide-treated bed nets [2] and indoor residual spraying [3], have proven effective but may not suffice when housing conditions are suboptimal [4]. However, challenges such as low bed nets adherence [5], rising insecticide resistance [6], and the structural unsuitability of IRS in poorly enclosed homes [7] have limited the effectiveness of these traditional tools in many high-transmission areas. Renovating homes with netting covering ceilings has the potential to further reduce Anopheles entry and resting, thereby lowering mosquito density indoors and around homes [8]. However, high-quality evidence on the effectiveness of such structural renovations in resource-limited settings is scarce.

Housing modifications have emerged as a promising avenue for malaria prevention. Improving housing infrastructure to reduce human exposure to mosquitoes has been associated with decreased malaria risk [9]. For instance, screening windows, doors, eaves, and/or ceilings can significantly limit mosquito entry into homes, thereby reducing indoor mosquito densities and subsequent malaria transmission [10]. A systematic review and meta-analysis concluded that better housing conditions are associated with a lower risk of malaria, underscoring the potential of housing improvements as a sustainable intervention [11]. Building upon this concept, the integration of long-lasting insecticide-treated netting (LLIN) into housing ceilings offers an innovative approach to vector control [12, 13]. LLINs have been extensively used as bed nets, providing both a physical barrier and an insecticidal effect against mosquitoes. Their application in housing modifications, such as installing either LLINs or mosquito nets across ceilings and open structural areas, aims to extend protection beyond sleeping spaces to the entire household environment. This strategy not only prevents mosquito entry but also targets mosquitoes that come into contact with the treated surfaces, potentially reducing both indoor and outdoor mosquito populations, with possible spillover effects on community-level mosquito abundance [9,14,15].

The stepped wedge cluster randomized trial (SW-CRT) design is particularly suited for evaluating interventions such as house renovations for malaria control. In an SW-CRT, all clusters eventually receive the intervention, but the implementation is staggered over time [16]. This design allows for the assessment of intervention effects while accounting for temporal changes and is often employed when the intervention is expected to do more good than harm, and it is logistically or ethically challenging to withhold it from any group [16]. The SW-RCT design has been effectively utilized in various public health studies, including those assessing malaria interventions [17,18]. In this context, we conducted a SW-RCT to evaluate the effectiveness

of installing ceiling-mounted mosquito nets (CMMNs) in reducing Anopheles mosquito density both indoors and peridomestic in the Llanchama community, San Juan Bautista district, Loreto, Peru. This study aims to contribute to the growing body of evidence supporting housing modifications as a viable strategy for malaria control and to explore the potential of mosquito nets beyond their traditional use as bed nets.

## Materials and methods

### Ethical statement

In accordance with international ethical standards and good research practices, the study protocol (S1 File) was reviewed and approved by the Ethics Review Board of the Iquitos Regional Hospital (Comité Institucional de Ética en Investigación del Hospital Regional de Loreto, Approval Certificate No. 08-CIEI-HRL-2024). Prior to any study-related activity, the legal head of each eligible household was approached by trained field staff and provided with detailed information about the study objectives, procedures, potential risks, and benefits. Written informed consent was obtained from all participating household heads. In cases involving participants with limited literacy, the consent form was read aloud in the presence of an independent literate witness, and consent was documented as per ethics committee guidelines.

### Study design

We conducted an open SW-RCT with four-steps, randomized at the household level. Using this approach, we invited the head of every eligible house to participate in the study as part of one the household clusters. The intervention was remodeling their houses to install CMMNs and the study outcomes were standard entomological indexes measured by human landing catching. Briefly, we planned the study including three clusters of Llanchama households randomly selected, which were intervened sequentially in four steps: (i) step 1 or baseline without intervention; (ii) step 2 or 3-month phase with the first cluster; (iii) step 3 or 6-month phase with the first and second cluster intervened; (iv) step 4 or 9-month phase with the tree clusters intervened (Table 1).

### Study population

The study was conducted in Llanchama, a rural village within the Zungarococha community near Iquitos, Peru—an area highly endemic for both *Plasmodium falciparum* and *P. vivax* malaria. Llanchama (total population ≈350) and its neighboring villages are served by a single Ministry of Health (Ministerio de Salud del Perú, MoH) post and are primarily accessed by unpaved roads with variable seasonal accessibility. Malaria surveillance and treatment in the region are carried out by MoH personnel through passive case detection and microscopy-based diagnosis, with standard antimalarial therapy provided at no cost. Households in Llanchama typically lack window screens, are closely spaced, and are situated near both natural water sources—such as rivers, small lakes, and seasonal flood zones—and artificial breeding sites, including fish

**Table 1. Stepped wedge cluster randomized trial design.**

| Cluster | Step 1 (Baseline) | Step 2 (3rd month) | Step 3 (6th month) | Step 4 (9th month) |
|---|---|---|---|---|
| Cluster 1 | 19 | 19 | 19 | 19 |
| Cluster 2 | 19 | 19 | 19 | 19 |
| Cluster 3 | 19 | 19 | 19 | 19 |
| Total | 57 (57 vs 0) | 57 (38 vs 19) | 57 (19 vs 38) | 57 (0 vs 57) |

farms and rainwater accumulation on the ground, all of which contribute to increased mosquito exposure. The community relies on agriculture and fishing, and most activities are performed outdoors or in unscreened environments due to the intermittent access to electricity, which limits indoor lighting and ventilation (S2 File).

## Study intervention

The intervention involved structurally modifying houses to install CMMNs and other open areas. This approach was designed to reduce mosquito entry, lower indoor Anopheles densities, and subsequently decrease malaria transmission risk. The remodeling process entailed retrofitting existing homes with durable, mosquito nets to create both a physical barrier against mosquitoes. Prior to remodeling, households were surveyed to document their structural characteristics, existing protective measures (e.g., bed nets, screens), and baseline entomological indicators. The intervention was tailored to the typical housing structures found in the Peruvian Amazon, which often feature elevated wooden floors, open eaves, and palm-thatched or corrugated metal roofs. Homes with large cracks or exposed sections in wooden plank walls had these areas covered with mosquito netting to further reduce entry points (14 houses received this modification). The netting material used was standard high-density polyethylene (HDPE) mosquito netting (0.28 monofilament with a fine mesh of ≥156 holes per square inch), designed to withstand humidity, ultraviolet exposure (UV protection of 600 kilo-light-years, equivalent to 4 years), and mechanical wear, thereby ensuring long-term protection (S2 File). The intervention was introduced in a phased manner across three clusters over nine months. Every three months, a new cluster of households received the materials and underwent remodeling, allowing for a controlled evaluation of the intervention's impact while ensuring that all households ultimately benefited. All houses in a cluster were remodeled within 3–5 days.

## Study outcomes

The study assessed the effectiveness of CMMNs in reducing mosquito density and human exposure, using standardized entomological methods. The primary outcomes were counts of Anopheles mosquitoes—captured indoors and peridomestic via human landing catches (HLCs)—to quantify overall mosquito abundance and spatial distribution within the community (S2 File). Secondary outcomes focused on malaria transmission risk, measured as bites per person per night (BPN) and bites per person per hour (BPH), both indoors and peridomestic. Mosquito collections followed standard methods [19] and involved oral aspirators, collection jars, and flashlights. Briefly, teams collected mosquitoes indoors and peridomestic (approximately 10 meters from the front door) over 12-hour periods (18:00–06:00) on two consecutive nights, rotating collectors every two hours to account for individual differences in attractiveness to mosquitoes. Data from these 12-hour HLC sessions were used to calculate BPN and BPH. The study outcomes were measured during the first week of each study steps (step 1 or baseline, step 2 or 3-month phase, step 3 or 6-month phase, and step 4 or 4-month phase). By comparing these entomological indicators before and after the intervention—and adjusting for the SW-RCT design—the study aimed to determine whether this structural modification effectively reduced the indoor presence of mosquitoes and, consequently, the potential for malaria transmission in a high-burden Amazonian community. No malaria case analysis was performed as this was an entomological efficacy trial.

## Study procedures

The study followed a structured series of procedures to ensure the rigorous implementation and evaluation of the intervention using a SW-CRT design. This approach allowed for a phased introduction of the intervention while ensuring that all households eventually benefited, enabling comparisons between intervention and control periods within the same community. Briefly, a total of 69 households were recruited from the Llanchama community in the San Juan Bautista district, Loreto, Peru, based on predefined eligibility criteria. Households were required to be permanent residents, structurally suitable for the intervention (e.g., presence of a roof structure for netting installation), and willing to participate in the full duration of the study. After recruitment, households were randomly assigned into three clusters, based on geographic

proximity and housing stratification, each receiving the intervention at different time points. The staggered implementation occurred in three-month intervals over the nine-month study period (March–December 2024), ensuring that all clusters served as controls before receiving the intervention.

Before the intervention was introduced, baseline entomological assessments were conducted in all 69 households to determine initial mosquito density and human exposure to bites. HLCs were performed indoors and peridomestic to estimate mosquito biting rates. Additionally, household characteristics such as roofing materials, wall structures, and existing malaria prevention practices (e.g., bed net usage) were documented. Community members were provided with detailed information about the study objectives, and informed consent was obtained from all participating households before data collection began.

During the intervention, the installation of ceiling-mounted mosquito nets was carried out by community residents themselves under the supervision of the project team. The nets were attached to ceiling structures and open eaves—common mosquito entry points in traditional Amazonian homes—creating a physical barrier to prevent mosquito entry. After installation, households received training on mosquito nets maintenance, including proper cleaning and handling to preserve the integrity and efficacy of the netting over time.

Post-Intervention, trained field teams conducted entomological monitoring to assess changes in mosquito populations and biting rates. Mosquito collections were performed both indoors and peridomestic using standardized entomological methods. HLCs continued to be conducted at fixed intervals to quantify mosquito BPN and BPH, both indoors and peridomestic. Environmental variables such as temperature and relative humidity were also recorded throughout the study to control for seasonal variations in mosquito activity that might influence results.

## Sample size

We used a weighted sampling approach to ensure that each study sequence included house types B, C, and D. We initially excluded type A houses because of their rudimentary structure and lack of permanent residents. From our mapping, we observed that the distribution of house types A, B, C, and D was approximately 10%, 20%, 30%, and 40%, respectively. We based our sample size calculation on an 80% study power, a 5% alpha level, and an expected 50% reduction in *Anopheles darlingi* indoor biting. Under these assumptions, we determined that a minimum of 57 houses would be needed for a SW-RCT design with staggered groups (19 houses per sequence), including one baseline and three follow-up periods. On the basis of this estimate, we decided to enroll 23 households per sequence, to account for a < 20% loss to follow-up due to any contingency, including abandoned households or households that did not complete the renovation or do so incorrectly.

A total of 83 households were assessed for eligibility, and 14 were excluded for not meeting inclusion criteria or having been abandoned (Fig 1). The remaining 69 households were randomly assigned to three sequences of 23 households each. Although all sequences proceeded through the SW-RCT, several households did not complete renovations according to the intended timeline and were excluded from subsequent follow-ups. In the end, each sequence concluded with 19 households, mirroring the situation in which certain groups, as in the example, transferred to a later sequence or opted out of the intervention cycles (Table 1).

## Randomization and allocation concealment

To ensure balance in structural housing characteristics across intervention arms, eligible households were first stratified by housing type (categories B, C, and D), based on pre-specified architectural features relevant to mosquito exposure and intervention performance (S2 File). Within each stratum, the study statistician—who was independent of field implementation—used the RAND() function in Microsoft Excel to generate a computer-based simple random allocation of households to one of the three stepped-wedge intervention sequences, assigning 23 households per sequence. The average distance between clusters was approximately 300–500 meters, sufficient to minimize contamination between groups given the short nightly flight range of *Anopheles darlingi* in this region (typically <1 km). Allocation was concealed from field teams

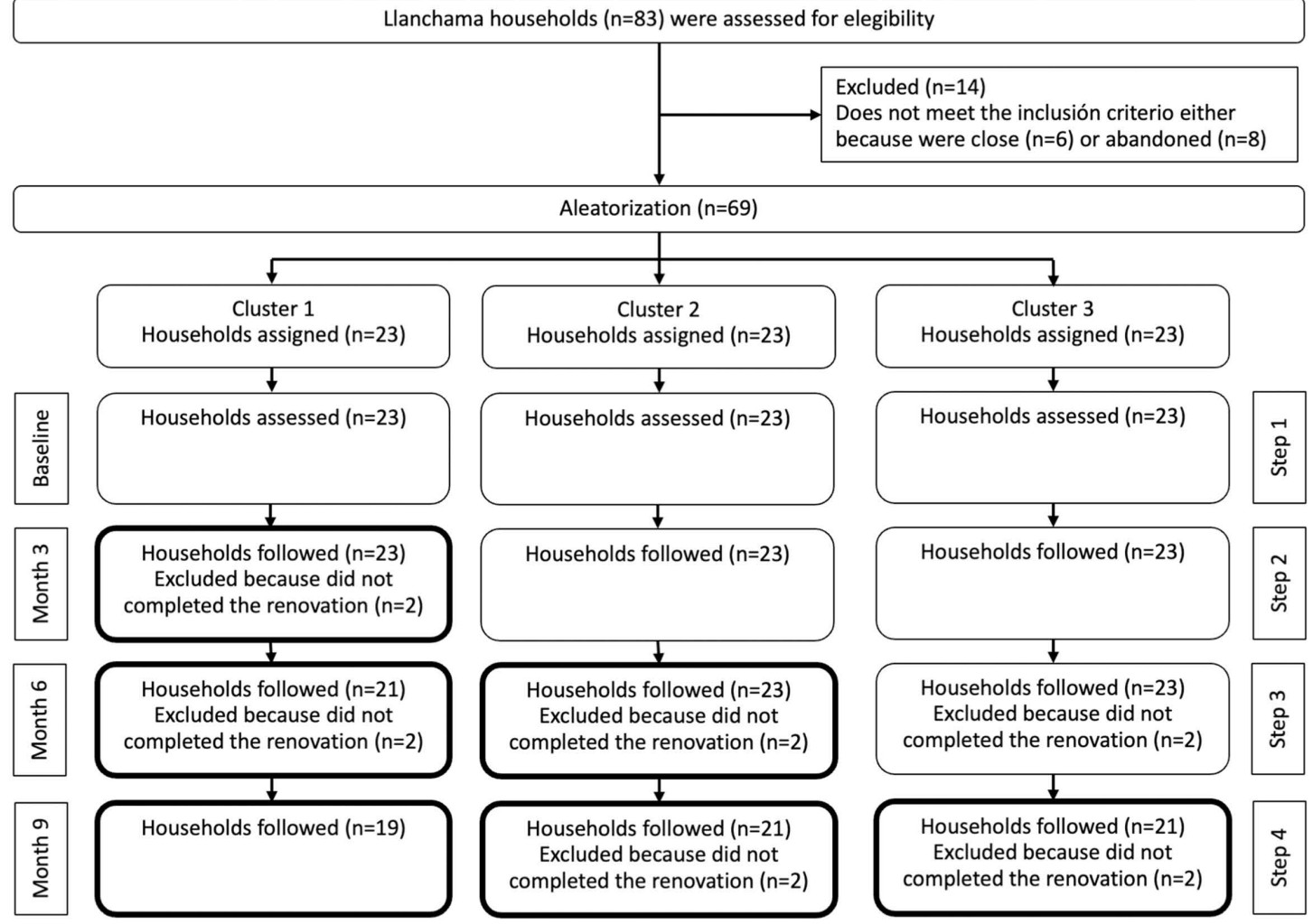

**Fig 1. Flow chart of the households (clusters) intervention.** The figure shows that a total of 83 households were assessed for eligibility; 14 were excluded, and 69 were randomly allocated to three sequences of 23 households each. Some households did not complete the renovation process within the allotted steps, resulting in exclusions from subsequent follow-ups. The chart illustrates the progression of each sequence through the baseline and intervention steps, and the final number of households analyzed per sequence (n = 57, 19 households per sequence).

and participants during the stratification and randomization process. Household assignments were disclosed only shortly before the initiation of their respective intervention phase, ensuring that neither participants nor intervention staff could influence the allocation. To minimize contamination between sequences, households were explicitly instructed not to share information, photos, or materials related to the intervention with neighbors in future sequences. Additionally, physical access to intervention materials was restricted until each household had completed its remodeling phase. These measures were reinforced through regular community meetings and home visits by blinded monitoring personnel.

## Data collection and analysis

Data were collected systematically during both the baseline and post-intervention phases to evaluate the effectiveness of the intervention in reducing Anopheles mosquito density and human-vector contact. Entomological outcomes included

indoor and peridomestic mosquito counts, as well as human biting rates, measured using standardized trapping protocols and exposure assessments conducted at consistent intervals across all clusters. The primary analysis followed an intention-to-treat (ITT) approach, whereby all households were analyzed according to their originally assigned intervention sequence, regardless of deviations in intervention timing or adherence. This approach preserved the integrity of the randomization and ensured valid between-group comparisons. To estimate the intervention's effect, we employed the Hussey and Hughes analytic framework for stepped-wedge cluster randomized trials, applying mixed-effects negative binomial regression models tailored to each entomological outcome [20]. These models accounted for within-cluster correlation and overdispersion in mosquito count data through the inclusion of: Random intercepts for cluster (household) to control for intra-household correlation; Fixed effects for time period to account for secular trends; Covariate adjustment for potential confounders, including housing type, environmental factors, and seasonal variation. Estimates of intervention effectiveness were expressed as adjusted incidence rate ratios (aIRRs) with corresponding 95% confidence intervals (CIs). This modeling strategy allowed for robust estimation of relative changes in mosquito density and biting rates associated with the intervention. We performed all statistical analyses using STATA/MP version 14.0 (StataCorp, College Station, TX, USA).

## Results

At baseline (from 03/15/24 to 03/25/24), median indoor Anopheles mosquito counts across the three clusters ranged from 36 to 62 mosquitoes per household. After the intervention, indoor mosquito counts decreased significantly, with median values dropping to 0–2 mosquitoes per household by the 9th month (Fig 2). Peridomestic mosquito counts were more variable, with baseline medians ranging from 63 to 79 mosquitoes per household, but showed no consistent reductions over time following CMMNs installation. Regarding human exposure to mosquito bites, indoor bites per person per night (BPN) were substantially reduced, from 18–31 at baseline to 0–1 by the 9th month. Peridomestic BPN, however, showed minimal change, remaining between 6 and 10 throughout the study period. Indoor bites per person per hour (BPH) followed a similar trend, with baseline values of 3–5 dropping to 0 after intervention, while peridomestic BPH remained relatively stable (Table 2).

According to the multivariable mixed-effects negative binomial regression models the intervention was associated with a 55% reduction in indoor mosquito counts (IRR: 0.45, 95% CI: 0.26–0.77, p = 0.004) but did not significantly reduce peridomestic mosquito counts (IRR: 0.94, 95% CI: 0.75–1.17, p = 0.570). Similarly, indoor BPN decreased by 60% (IRR: 0.40, 95% CI: 0.22–0.73, p = 0.003), while peridomestic BPN remained unchanged (IRR: 0.93, 95% CI: 0.73–1.19, p = 0.56). Hourly bite rates (BPH) followed the same pattern, with indoor BPH decreasing by 61% (IRR: 0.39, 95% CI: 0.17–0.85, p = 0.018), whereas peridomestic BPH was not significantly affected (IRR: 0.92, 95% CI: 0.57–1.49, p = 0.740). Temporal analysis demonstrated that mosquito densities and biting rates declined sharply after the 3rd month, with greater reductions at the 6th and 9th months. By the 9th month, indoor mosquito counts were 81% lower than at baseline (IRR: 0.19, 95% CI: 0.14–0.26, p < 0.001), and indoor BPN was reduced by 88% (IRR: 0.12, 95% CI: 0.05–0.26, p < 0.001) (Table 3). Any harm was reported during the study.

## Discussion

This SW-CRT conducted in a malaria-endemic, riverine community of the Peruvian Amazon provides robust evidence that CMMNs are highly effective at reducing indoor Anopheles mosquito exposure. Specifically, we observed statistically significant reductions of 55% in indoor mosquito density, 60% in indoor BPN, and 61% in BPH following intervention deployment. These effects were temporally consistent and sustained across three sequential clusters, reinforcing the durability of the intervention effect. Notably, peridomestic mosquito exposure did not decline significantly, underscoring the spatial specificity of the protective benefit conferred by CMMNs.

Unlike traditional vector control tools such as long-lasting insecticide-treated nets (LLINs) or indoor residual spraying (IRS), CMMNs offer a passive, infrastructural solution that does not depend on daily behavioral adherence. As such, they

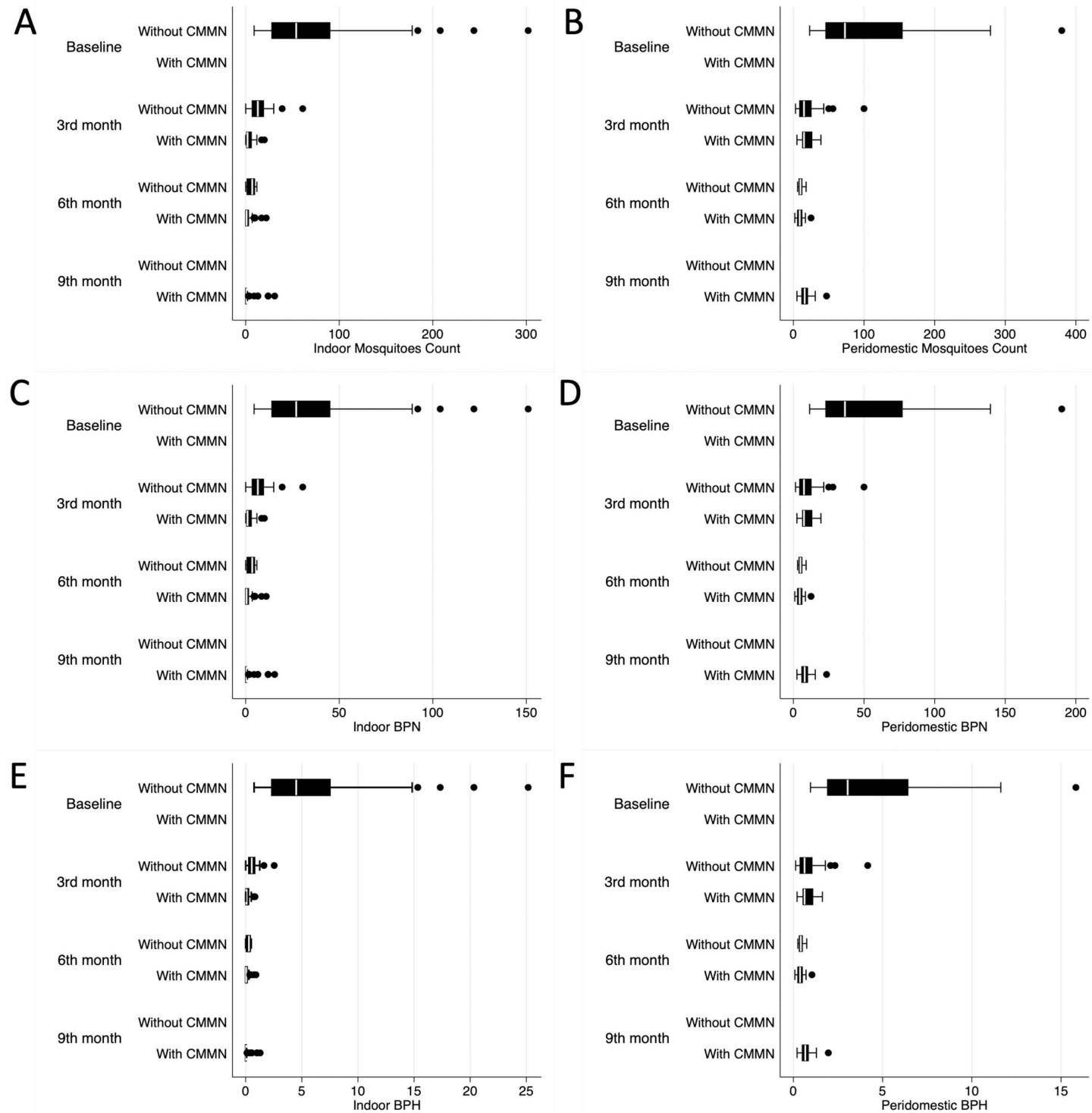

**Fig 2. Mosquito density and biting rates during the study.** The figure shows the monitoring of the mosquito density and biting rates during each study phase, including indoor mosquitoes counts (Panel 2A), peridomestic mosquitoes count (Panel 2B), indoor BPN (Panel 2C), peridomestic BPN (Panel 2D), indoor BPH (Panel 2E), and peridomestic BPH (Panel 2F). CMMN, ceiling-mounted mosquito nets; BPN, bites per person per night; BPN, bites per person per person; Gray shadow, intervention period.

**Table 2. Mosquito density and biting rates during the study.**

| Cluster | Entomological index | Baseline | 3rd month | 6th month | 9th month |
|---|---|---|---|---|---|
| | | Median (IQR) | Median (IQR) | Median (IQR) | Median (IQR) |
| 1 | Indoor mosquitoes count | 62 (38) | 2 (5) | 1 (2) | 1 (1) |
| | Peridomestic mosquitoes count | 79 (82) | 15 (13) | 7 (6) | 16 (8) |
| | Total count | 136 (128) | 22 (16) | 7 (6) | 17 (8) |
| | Indoor BPN | 31 (19) | 1 (3) | 1 (1) | 1 (1) |
| | Peridomestic BPN | 40 (41) | 8 (7) | 3 (3) | 8 (4) |
| | Overall BPN | 68 (64) | 11 (8) | 4 (3) | 8 (4) |
| | Indoor BPH | 5 (3) | 0 (0) | 0 (0) | 0 (0) |
| | Peridomestic BPH | 3 (3) | 1 (1) | 0 (0) | 1 (0) |
| | Overall BPH | 6 (5) | 1 (1) | 0 (0) | 1 (0) |
| 2 | Indoor mosquitoes count | 36 (68) | 11 (8) | 2 (9) | 0 (1) |
| | Peridomestic mosquitoes count | 71 (105) | 14 (15) | 11 (4) | 19 (10) |
| | Total count | 122 (174) | 25 (19) | 15 (8) | 19 (11) |
| | Indoor BPN | 18 (34) | 6 (4) | 1 (5) | 0 (1) |
| | Peridomestic BPN | 36 (53) | 7 (8) | 6 (2) | 10 (5) |
| | Overall BPN | 61 (87) | 13 (10) | 8 (4) | 10 (6) |
| | Indoor BPH | 3 (6) | 0 (0) | 0 (0) | 0 (0) |
| | Peridomestic BPH | 3 (4) | 1 (1) | 0 (0) | 1 (0) |
| | Overall BPH | 5 (7) | 1 (1) | 1 (0) | 1 (0) |
| 3 | Indoor mosquitoes count | 60 (74) | 17 (10) | 7 (8) | 2 (7) |
| | Peridomestic mosquitoes count | 63 (192) | 18 (24) | 10 (4) | 15 (5) |
| | Total count | 105 (252) | 38 (20) | 18 (6) | 18 (12) |
| | Indoor BPN | 30 (37) | 8 (5) | 4 (4) | 1 (3) |
| | Peridomestic BPN | 32 (96) | 9 (12) | 5 (2) | 7 (2) |
| | Overall BPN | 52 (126) | 19 (10) | 9 (3) | 9 (6) |
| | Indoor BPH | 5 (6) | 1 (0) | 0 (0) | 0 (0) |
| | Peridomestic BPH | 3 (8) | 1 (1) | 0 (0) | 1 (0) |
| | Overall BPH | 4 (10) | 2 (1) | 1 (0) | 1 (1) |

* IQR, interquartile range; BPN, bites per person per night; BPN, bites per person per person; Gray shadow, intervention period.

represent a promising addition to the malaria prevention toolkit, particularly in settings where nightly net use is suboptimal or household structures remain highly permeable to vector entry.

Our findings are concordant with an emerging consensus that housing quality is a major determinant of malaria risk. Multiple systematic reviews and meta-analyses have affirmed the protective role of improved housing design, independent of LLIN or IRS coverage. Tusting et al. reported a 47% relative reduction in malaria infection associated with interventions that physically block mosquito entry, such as screening and improved eaves [9]. Similarly, a meta-analysis by Kua and Lee concluded that screened homes were associated with a 37% lower malaria incidence, supporting a shift toward structural, non-chemical interventions [11].

The magnitude of indoor vector suppression in our trial aligns with data from other housing intervention studies. Trials conducted in sub-Saharan Africa have shown that closing eaves and installing window screening significantly reduces indoor mosquito density and malaria incidence [9–11]. For instance, a trial in rural Gambia observed a 65% reduction in Anopheles exposure following house modifications [8]. However, few studies to date have specifically targeted ceiling areas, despite their critical role in vector entry in elevated, open-eave homes typical of Amazonian riverine environments.

**Table 3. Estimated Effects of CMMNs on Mosquito Density and Biting Rates.**

| Model Parameters | Indoor mosquitoes count | Peridomestic mosquitoes count | Overall mosquitoes count | Indoor BPN | Peridomestic BPN | Overall BPN | Indoor BPH | Peridomestic BPH | Overall BPH |
|---|---|---|---|---|---|---|---|---|---|
| | IRR (95% CI) | IRR (95% CI) | IRR (95% CI) | IRR (95% CI) | IRR (95% CI) | IRR (95% CI) | IRR (95% CI) | IRR (95% CI) | IRR (95% CI) |
| CMMN | **0.45** (0.26–0.77)** | 0.94ns (0.75–1.17) | 0.78ns (0.59–1.03) | **0.40** (0.22–0.73)** | 0.93ns (0.73–1.19) | 0.75ns (0.55–1.02) | **0.39* (0.17–0.85)** | 0.92ns (0.57–1.49) | 0.71ns (0.47–1.08) |
| Time | | | | | | | | | |
| Baseline | Reference | Reference | Reference | Reference | Reference | Reference | Reference | Reference | Reference |
| 3rd month | 0.20*** (0.14–0.30) | 0.20*** (0.16–0.25) | 0.20*** (0.16–0.24) | 0.21*** (0.14–0.30) | 0.20*** (0.16–0.25) | 0.20*** (0.16–0.25) | 0.10*** (0.07–0.16) | 0.20*** (0.14–0.28) | 0.20*** (0.15–0.27) |
| 6th month | 0.09*** (0.05–0.16) | 0.10*** (0.08–0.13) | 0.10*** (0.07–0.13) | 0.10*** (0.06–0.18) | 0.10*** (0.08–0.13) | 0.10*** (0.07–0.13) | 0.05*** (0.02–0.11) | 0.10*** (0.06–0.18) | 0.10*** (0.07–0.16) |
| 9th month | 0.45*** (0.26–0.77) | 0.19*** (0.14–0.26) | 0.17*** (0.12–0.24) | 0.12*** (0.05–0.26) | 0.19*** (0.14–0.27) | 0.17*** (0.11–0.26) | 0.06*** (0.02–0.20) | 0.21*** (0.11–0.39) | 0.20*** (0.11–0.35) |
| Household type | | | | | | | | | |
| Type B | Reference | Reference | Reference | Reference | Reference | Reference | Reference | Reference | Reference |
| Type C | 1.07 ns (0.66–1.73) | 0.88ns (0.67–1.15) | 0.92ns (0.71–1.19) | 1.04ns (0.67–1.61) | 0.86ns (0.65–1.13) | 0.91ns (0.70–1.18) | 0.84ns (0.56–1.27) | 0.73ns (0.53–1.02) | 0.79ns (0.59–1.07) |
| Type D | 0.45** (0.26–0.77) | 0.76ns (0.67–1.03) | 0.69* (0.52–0.92) | 0.46** (0.29–0.73) | 0.73* (0.53–0.99) | 0.68** (0.51–0.91) | 0.51** (0.31–0.85) | 0.56** (0.37–0.83) | 0.56** (0.39–0.80) |
| Constant | 122.0*** (94.4–157.5) | 122.0*** (94.4–157.5) | 197.5*** (153.6–254.0) | 39.6*** (26.0–60.5) | 62.6*** (48.1–80.3) | 100.0*** (78.0–128.1) | 7.3*** (5.0–10.6) | 6.2*** (4.5–8.2) | 9.5*** (7.2–12.5) |
| Ln(alpha) | −0.14*** (−0.36–0.08) | −1.26*** (−1.45 − −1.07) | −1.36*** (−1.55 − −1.17) | −0.43** (−0.68 − −0.17) | −1.32*** (−1.53 − −1.11) | −1.41*** (−1.62 − −1.20) | −1.34*** (−1.82 − −0.86) | −2.30*** (−3.10 − −1.51) | −2.04*** (−2.56 − −1.52) |
| Cluster Variance | 0.02*** (<0.01–0.52) | <0.01NA (NA) | <0.01NA (<0.01–0.21) | <0.01NA (NA) | <0.01 (NA)NA | <0.01NA (<0.01–1.27) | <0.01NA (NA) | <0.01NA (NA) | <0.01NA (NA) |

*, p value <0.05; **, p value <0.01; ***, p value <0.001; ns, non-significant; IRR, incidence rate ratio; BPN, bites per person per night; BPN, bites per person per person; Ln, Napierian logarithm; CMMN, ceiling-mounted mosquito nets.

Notably, our findings extend the work of Kagaya et al. and Ko et al., who evaluated the use of insecticidal ceiling nets in Kenyan children through cluster randomized trials. Although those studies focused on malaria incidence, their mechanistic premise—blocking upward mosquito entry—was conceptually validated by our entomological outcomes [12,13]. Furthermore, although our study primarily targeted Anopheles vectors, incidental reductions in other mosquito genera were observed during the study period, particularly indoors. Structural modifications such as ceiling netting likely disrupt entry pathways for both malaria and non-malaria vectors. This is supported by previous research demonstrating that improved housing design, screened ceilings, and closed eaves reduce indoor densities of Culex and Aedes mosquitoes, contributing to broader vector control benefits [21,22]. Future studies should quantify the effect of CMMNs on non-Anopheles species to assess their potential for integrated vector management.

Moreover, our study reinforces evidence from SW-CRTs evaluating vector control interventions. The SolarMal trial in Kenya, which used mass mosquito trapping, demonstrated a significant reduction in malaria prevalence, validating the SW-CRT design for evaluating community-level interventions under pragmatic conditions [17]. Likewise, trials of repellents

delivered via community health workers in Myanmar showed the feasibility and operational utility of staged implementation designs in high-burden settings [18].

The observed lack of peridomestic efficacy highlights a persistent challenge in malaria elimination: residual transmission driven by exophagic and exophilic vectors [19,23]. In Amazonian settings, *Anopheles darlingi* exhibits predominantly endophagic behavior [24], which likely explains the intervention's indoor-specific effect. However, residual transmission persists outdoors, especially in early evening and peri-domestic contexts [25,26], pointing to the need for complementary interventions targeting exophilic behavior—such as spatial repellents [27], insecticide-treated clothing [28], and attractive targeted sugar baits (ATSBs) [29].

Interestingly, we observed variation in intervention efficacy by housing type. Type D homes—typically more enclosed—demonstrated greater relative reductions in vector metrics compared to Types B and C. This heterogeneity suggests that baseline housing permeability may act as an effect modifier, a hypothesis supported by prior research in Uganda, where housing configuration modulated the protective effects of LLINs and IRS [4]. Such heterogeneity underscores the need to tailor structural interventions to specific housing archetypes and supports stratified analysis in future trials.

From a health systems perspective, the potential scalability of CMMNs is notable. As a one-time installation with low maintenance requirements, the intervention circumvents common barriers to behavioral compliance that undermine other malaria control strategies. Given the logistical and financial challenges of sustained LLIN and IRS coverage in remote, forested settings, CMMNs may offer a durable and acceptable complement for national programs, particularly in hard-to-reach populations [3].

This study's principal strength lies in its pragmatic SW-CRT design, which enabled robust causal inference while ensuring equitable delivery of the intervention across all clusters. Within-cluster comparisons minimized temporal confounding, and the use of mixed-effects negative binomial models accounted for clustering, period effects, and structural heterogeneity among households. The use of human landing catches—considered the gold standard for measuring human-vector contact—strengthened outcome measurement fidelity. Nonetheless, several limitations should be considered. First, our reliance on entomological proxies—mosquito density and human biting rates—precludes direct conclusions about the intervention's effect on malaria infection or disease, although these metrics are strongly associated with transmission risk. Second, the lack of blinding may have introduced performance or detection bias; however, the objective nature of entomological outcomes and standardized protocols likely reduced this risk. Third, contamination or spillover effects between clusters—due to geographic proximity and social interactions—may have attenuated contrasts between intervention sequences. While the average separation between clusters was 300–500 meters, and households were grouped to minimize inter-cluster proximity, it is possible that mosquitoes deflected by treated homes migrated to untreated ones in neighboring clusters. Importantly, studies in the Peruvian Amazon indicate that its typical host-seeking range is much shorter (often <500 meters) [19], and its movement is generally constrained by environmental features and host availability. Furthermore, our use of cluster-level modeling with random intercepts helped mitigate this bias, though some residual inter-cluster influence may persist. Fourth, although we adjusted for key environmental covariates, residual confounding from unmeasured or dynamic ecological factors (e.g., microclimatic variation or larval habitat changes) cannot be ruled out. Finally, clinical malaria outcomes were not assessed due to logistical and resource limitations. As a result, our evaluation focused exclusively on entomological indicators of transmission risk. Future studies should integrate parasitological outcomes, such as confirmed malaria cases or infection prevalence, alongside entomological data to more comprehensively assess the epidemiological impact of structural interventions like ceiling-mounted mosquito nets.

Following CONSORT guidelines for SW-CRTs [30], these findings should be interpreted in light of the study's implementation context. While our results suggest entomological efficacy in a rural Amazonian community, generalizability may be limited to areas with similar housing designs, vector ecology, and health system capacities. Future studies should incorporate clinical or parasitological endpoints such as malaria incidence, PCR-confirmed parasitemia, or test positivity rates, and assess cost-effectiveness under real-world programmatic conditions. Evaluations comparing treated versus

untreated netting are needed to clarify the incremental benefit of insecticidal properties in areas of evolving insecticide resistance. Additionally, operational research should explore the durability, household maintenance practices, and long-term acceptability of CMMNs, especially when integrated into comprehensive housing improvement strategies. Finally, mechanistic entomological studies, including vector behavior assessments, resting patterns, and escape responses, are warranted to refine intervention specifications and inform optimal deployment strategies for different household typologies and transmission settings.

## Conclusions

This SW-CRT provides compelling evidence that CMMNs significantly reduce indoor Anopheles mosquito exposure in high-transmission, structurally vulnerable communities of the Peruvian Amazon. In an era where malaria control increasingly requires integrated, sustainable, and context-adapted strategies, structural interventions such as CMMNs deserve greater inclusion in the global vector control agenda. By offering high community acceptability, low behavioral dependency, and strong ecological compatibility, CMMNs represent a scalable and equitable complement to conventional interventions—particularly in populations underserved by current vector control tools. As the global health community strives toward malaria elimination, investments in durable, infrastructure-based solutions like CMMNs may be critical for reaching residual transmission foci and achieving lasting impact.

## Supporting information

**S1 File. Ethics approval certificate and detailed study protocol.** This file includes the official ethics approval certificate (Certificate No. 08-CIEI-HRL-2024) issued by the Loreto Regional Hospital Ethics Committee and the complete protocol for the stepped-wedge cluster randomized trial evaluating ceiling-mounted mosquito nets in the Llanchama community. The protocol details study objectives, design, sampling strategy, data collection and analysis methods, ethical considerations, and includes informed consent templates for participants and field collectors.
(PDF)

**S2 File. Study population, geographic context, and intervention setting.** This file provides detailed information on the study site and population characteristics in Llanchama, a rural Amazonian village in Loreto, Peru. It includes maps and drone images of the community layout, descriptions of local socioeconomic and environmental conditions, malaria transmission dynamics, and housing features relevant to vector exposure. The document also summarizes the implementation of CMMNs within the stepped-wedge cluster randomized trial, including structural modifications to typical Amazonian homes and baseline entomological context.
(PDF)

## Acknowledgments

We thank the residents of Llanchama, as well as the GERESA staff, volunteers, and field workers, for their diligent efforts in documenting the study's entomological outcomes.

## Author contributions

**Conceptualization:** Antonio Marty Quispe, Cinthia Inti, Eun Seok Kim.

**Data curation:** Antonio Marty Quispe, Carlos Álvarez-Antonio, Freddy Gutierrez, Cristiam Carey.

**Formal analysis:** Antonio Marty Quispe.

**Funding acquisition:** Cristiam Carey, Hugo Rodríguez-Ferrucci, Cinthia Inti, Eun Seok Kim.

**Investigation:** Antonio Marty Quispe, Carlos Álvarez-Antonio, Freddy Gutierrez Rodriguez.

**Methodology:** Antonio Marty Quispe, Freddy Gutierrez Rodriguez.

**Project administration:** Hugo Rodríguez-Ferrucci, Cinthia Inti, Eun Seok Kim.

**Resources:** Carlos Álvarez-Antonio, Freddy Gutierrez Rodriguez, Cristiam Carey, Hugo Rodríguez-Ferrucci.

**Supervision:** Antonio Marty Quispe, Hugo Rodríguez-Ferrucci, Cinthia Inti, Eun Seok Kim.

**Validation:** Antonio Marty Quispe.

**Visualization:** Antonio Marty Quispe.

**Writing – original draft:** Antonio Marty Quispe.

**Writing – review & editing:** Antonio Marty Quispe, Carlos Álvarez-Antonio, Freddy Gutierrez Rodriguez, Cristiam Carey, Hugo Rodríguez-Ferrucci, Cinthia Inti, Eun Seok Kim.

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
