## [Decision Letter · Decision Letter 0]

11 Jul 2025

PONE-D-25-21398

Efficacy of Ceiling-Mounted Mosquito Nets for Malaria Vector Control in a Peruvian Amazon Riverine Community: A Stepped-Wedge Cluster Randomized Trial

PLOS ONE

Dear Dr.  Quispe,

Thank you for submitting your manuscript to PLOS ONE. After careful consideration, we feel that it has merit but does not fully meet PLOS ONE’s publication criteria as it currently stands. Therefore, we invite you to submit a revised version of the manuscript that addresses the points raised during the review process.

Specifically, while both reviewers felt the draft manuscript was strong and well-written, though Reviewer 1 felt like there were several areas that needed additional clarification before being further considered for publication, in particular the mosquito collection methodology in addition to several other areas throughout the manuscript. Reviewer 2 additionally felt like the draft could benefit from additional clarity and changes in the abstract.

Please ensure all these comments are fully addressed in any subsequent versions of the manuscript.

We look forward to receiving your revised manuscript.

Kind regards,

James Colborn

Academic Editor

PLOS ONE

Journal Requirements:

3. Please note that your Data Availability Statement is currently missing the repository name and/or the DOI/accession number of each dataset OR a direct link to access each database. If your manuscript is accepted for publication, you will be asked to provide these details on a very short timeline. We therefore suggest that you provide this information now, though we will not hold up the peer review process if you are unable.

4. Please include captions for your Supporting Information files at the end of your manuscript, and update any in-text citations to match accordingly. Please see our Supporting Information guidelines for more information: http://journals.plos.org/plosone/s/supporting-information .

5. We note that Figure S1 in your submission contain map/satellite images which may be copyrighted. All PLOS content is published under the Creative Commons Attribution License (CC BY 4.0), which means that the manuscript, images, and Supporting Information files will be freely available online, and any third party is permitted to access, download, copy, distribute, and use these materials in any way, even commercially, with proper attribution. For these reasons, we cannot publish previously copyrighted maps or satellite images created using proprietary data, such as Google software (Google Maps, Street View, and Earth). For more information, see our copyright guidelines: http://journals.plos.org/plosone/s/licenses-and-copyright.

 1. You may seek permission from the original copyright holder of Figure S1 to publish the content specifically under the CC BY 4.0 license. 

6. We note that Figure S3 includes an image of a participant in the study.

Reviewers' comments:

Reviewer's Responses to Questions

**Comments to the Author**

1. Is the manuscript technically sound, and do the data support the conclusions?

Reviewer #1: Yes

Reviewer #2: Yes

2. Has the statistical analysis been performed appropriately and rigorously? 

Reviewer #1: I Don't Know

Reviewer #2: Yes

3. Have the authors made all data underlying the findings in their manuscript fully available?

Reviewer #1: Yes

Reviewer #2: Yes

4. Is the manuscript presented in an intelligible fashion and written in standard English?

Reviewer #1: Yes

Reviewer #2: Yes

5. Review Comments to the Author

Reviewer #1: Review results

The authors aim to assess the effectiveness of ceiling-mounted mosquito nets in reducing Anopheles mosquito density both indoors and outdoors, with a focus on entomological aspects. General comments address the unclear procedural elements of this study, particularly concerning cluster allocation and the execution of HLC activities before, during, and after the intervention in the three study clusters.

ABSTRACT

1. Line 21-22: "We aimed to assess the effectiveness of ceiling-mounted mosquito nets in reducing Anopheles mosquito density both indoors and outdoors."

Based on the results obtained, it can be concluded that ceiling-mounted mosquito nets are only effective in indoor settings. How does the research objective describe the results in outdoor settings? It is recommended that the research objective avoid using the words 'indoors' and 'outdoors' (to assess the effectiveness of ceiling-mounted mosquito nets in reducing Anopheles mosquito density).

2. Line 26-29: "We measured entomological indices using human landing catches and other standardized methods, focusing on Anopheles mosquito counts, bites per person per night (BPN), and bites per person per hour (BPH), both indoors and outdoors."

Are these entomological endpoints ('mosquito counts', 'BPN', and 'BPH') analysed from the same HLC results? I think these variables have the same meaning.

3. Was there also an analysis comparing malaria cases in renovated and non-renovated households during the intervention periods?

INTRODUCTION

4. Line 67-69: "This strategy not only prevents mosquito entry but also targets mosquitoes that come into contact with the treated surfaces, potentially reducing both indoor and outdoor mosquito populations."

Please add references that support the statement, particularly those that address reducing the outdoor mosquito population.

STUDY DESIGN

5. Line 105-106: "Briefly, we planned the study including three clusters of Llanchama households randomly selected,"

What is the basis for the randomisation of clusters? Is it a comparison of house types (later referred to as types B, C, and D) within one cluster, or is it simply the number of houses grouped closely together at the study sites?

6. There are no marks or clear boundaries for each cluster on the study map (see Supplementary Material S2).

7. What is the distance between clusters?

8. If the clusters did not have clear boundaries, it would be possible for houses from different clusters to be very close to each other. Considering that Anopheles can fly quite long distances, would there be any bias if Anopheles could not enter a treated house, but could fly to an untreated house next door?

9. Line 111: Table 1 does not show how long it takes to install ceiling-mounted mosquito nets in households that received the intervention. Can installation be carried out in all households in a cluster simultaneously in a short time (less than one day)? Or did it take three months, including the installation process, for all households to ultimately benefit?

STUDY INTERVENTION

10. Lines 137–139: 'Homes with large cracks or exposed sections in wooden plank walls had these areas covered with mosquito netting to reduce entry points further.'

Are there any records of how many houses received additional improvements involving mosquito nets on the walls?

STUDY OUTCOMES

11. Line 152–153: 'Captured indoors and outdoors via human landing catches (HLCs).'

This is not explained in detail:

• How many times were HLCs conducted on two consecutive nights in each cluster, for example, once a week or biweekly?

• How many sentinel houses were there, and where were they located within the cluster?

STUDY PROCEDURES

12. Line 184–186: 'Before the intervention was introduced, baseline entomological assessments were conducted in all households to determine the initial density of mosquitoes and the extent of human exposure to bites.'

Did the HLCs conduct baseline entomological assessments in all 69 households?

SAMPLE SIZE

13. Line 210: What are the differences between house types A, B, C, and D? Could you please provide the classification criteria for house types A, B, C, and D, along with pictures if possible? You can include the explanation in Supplementary Section S2.

RESULTS

14. Lines 298–300: Outdoor BPN (Fig. 2D), indoor BPH (Fig. 2E), and outdoor *BPN (Fig. 2F). IQR should be BPH.

15. Line 299: Interquartile Range; *CMMN: Ceiling-mounted Mosquito Nets (does not refer to Figure 2).

16. Line 303–304: IQR (interquartile range); BPN (bites per person per night); *BPN (bites per person per hour); grey shadow (intervention period). (Should be 'BPH', bites per person per hour.)

DISCUSSION

17. The effect of CMMN on non-Anopheles mosquitoes is worth discussing, particularly given that the number of Anopheles mosquitoes decreased indoors and outdoors during the study period.

Reviewer #2: Thank you for the opportunity to review this manuscript. This is the good manuscript as it is portraying a new intervention that can be used to control mosquito-borne diseases especially indoor transmissions. Great work

Abstract

Statistical analysis (how the data were analyzed) is missing in the abstract section. It is important to appear in the abstract section

Line 40: I don’t think funding is part of the abstract. The authors should revise the journal submission guidelines if this adheres to these guidelines

Introduction section

Line 47-49 Authors have described the vector control tools such as ITNs and IRS. Could the authors describe the challenges that may be facilitating the persistence of malaria transmission despite the use of these traditional interventions?

The Introduction is well narrated

Materials and methods

The method section is well explained and it is sufficient. The study design, study procedures and data analysis section are well narrated. The authors have done a good job

Results

Line 265-280 should be moved to the methods section

Discussion

I have no comment in the discussion section, it is well written and it made the study very important.

6. PLOS authors have the option to publish the peer review history of their article (what does this mean? ). If published, this will include your full peer review and any attached files.

**Do you want your identity to be public for this peer review?** For information about this choice, including consent withdrawal, please see our Privacy Policy .

Reviewer #1: No

Reviewer #2: No

---

## [Author Response · Author response to Decision Letter 1]

25 Aug 2025

We sincerely thank the Academic Editor and both reviewers for their constructive and insightful feedback on our manuscript titled “Efficacy of Ceiling-Mounted Mosquito Nets for Malaria Vector Control in a Peruvian Amazon Riverine Community: A Stepped-Wedge Cluster Randomized Trial.” We have carefully revised the manuscript in accordance with the journal’s guidelines and addressed all comments point by point. These revisions have substantially improved the clarity, rigor, and overall quality of our work. Below, we detail our responses and the corresponding changes in the manuscript.

I. RESPONSES TO EDITORIAL REQUIREMENTS

Comment 1. Compliance with PLOS ONE style requirements (including file naming)

Editor’s Comment: Please ensure that your manuscript meets PLOS ONE's style requirements, including those for file naming.

Response:

Thank you for noting this. We have reviewed and formatted the manuscript according to PLOS ONE’s style templates. File names have been updated to conform to the required naming convention (e.g., `Manuscript.docx`, `Figure1.tif`, `SupportingInformation_S1.pdf`).

Comment 2. Removal of funding information from the manuscript text

Editor’s Comment: Funding information should not appear in the manuscript except in the Funding Statement in the online form.

Response:

We appreciate this clarification. We have removed all references to funding in the manuscript text, including the Abstract (previously line 40) and Acknowledgments. The funding statement is now only provided in the submission system as per PLOS ONE policy.

3. Complete Data Availability Statement

Editor’s Comment: The Data Availability Statement must include the repository name and DOI/accession number.

Response:

We have updated the Data Availability Statement (formerly “Data sharing” section) to include the repository name and DOI. The revised text (line 566–568) now reads:

“The anonymized, aggregated data collected for this study is available at Figshare under the Creative Commons Attribution 4.0 (CC BY 4.0) license: https://doi.org/10.6084/m9.figshare.28830449.”

Comment 4. Supporting Information captions

Editor’s Comment: Please include captions for your Supporting Information files at the end of your manuscript.

Response:

We have added complete captions for all Supporting Information files as requested (lines 113–119 and 149–156), including descriptions of their content and context. In-text citations have been updated accordingly.

Comment 5. Figure S1 contains potentially copyrighted maps

Editor’s Comment: Remove or replace copyrighted maps/images unless permission under CC BY 4.0 is obtained.

Response:

We could not obtain CC BY 4.0 permission for the original maps. Therefore, we replaced Figure S2A with a newly created map using OpenStreetMap (public domain data), which fully complies with CC BY 2.0 licensing. The figure caption has been updated to indicate its source.

Comment 6. Figure S3 includes a participant image

Editor’s Comment: Ensure informed consent for publication under CC BY license or remove the image.

Response:

We have removed the figure. Instead, we obtained CC BY 4.0 permission from World Vision Peru for figures S2B (Drone picture of the Llanchama community in the Loreto Region, Peru) and S2D (Ceiling-mounted mosquito nets at the Llanchama community), and from Blgo. Freddy Gutierrez Rodriguez for figure S2C (Household types A, B, C, and D at the Llanchama community).

Comment 7. Reviewer-requested citations

Editor’s Comment: Review and evaluate requested citations for relevance.

Response:

We assessed all suggested references and included relevant ones in the Introduction (e.g., \[5], \[6], \[7]) and Discussion (e.g., , \[19], \[21], \[22]) to strengthen the rationale for housing-based interventions and ceiling netting approaches (lines 55–58, 454–462).

II. RESPONSES TO REVIEWER #1 COMMENTS

ABSTRACT

Comment 1 (Line 21-22): "We aimed to assess the effectiveness of ceiling-mounted mosquito nets in reducing Anopheles mosquito density both indoors and outdoors." Based on the results obtained, it can be concluded that ceiling-mounted mosquito nets are only effective in indoor settings. How does the research objective describe the results in outdoor settings? It is recommended that the research objective avoid using the words 'indoors' and 'outdoors' (to assess the effectiveness of ceiling-mounted mosquito nets in reducing Anopheles mosquito density).

Response:

We agree. The sentence now reads (lines 21–22): “We aimed to assess the effectiveness of ceiling-mounted mosquito nets in reducing Anopheles mosquito density in a high-transmission Amazonian setting.”

Comment 2 (Line 26-29): "We measured entomological indices using human landing catches and other standardized methods, focusing on Anopheles mosquito counts, bites per person per night (BPN), and bites per person per hour (BPH), both indoors and outdoors." Are these entomological endpoints ('mosquito counts', 'BPN', and 'BPH') analyzed from the same HLC results? I think these variables have the same meaning.

Response:

Thank you for the observation. We have clarified this in the Methods (lines 27-30): “We measured entomological indices using human landing catches (HLC) and other standardized methods, focusing on Anopheles mosquito counts, bites per person per night (BPN), and bites per person per hour (BPH), both indoors and outdoors, derived from the same HLC sessions, using cumulative mosquito captures over 12-hour periods to compute biting indices.”

Comment 3: Was there also an analysis comparing malaria cases in renovated and non-renovated households during the intervention periods?

Response:

No malaria case analysis was performed as this was an entomological efficacy trial. We clarified this in the Methods and Discussion, stating:

- Line 201-202: “No malaria case analysis was performed as this was an entomological efficacy trial

- Lines 518-522: “Finally, clinical malaria outcomes were not assessed due to logistical and resource limitations. As a result, our evaluation focused exclusively on entomological indicators of transmission risk. Future studies should integrate parasitological outcomes, such as confirmed malaria cases or infection prevalence, alongside entomological data to more comprehensively assess the epidemiological impact of structural interventions like ceiling-mounted mosquito nets.”

INTRODUCTION

Comment 4 (Line 67-69): "This strategy not only prevents mosquito entry but also targets mosquitoes that come into contact with the treated surfaces, potentially reducing both indoor and outdoor mosquito populations." Please add references that support the statement, particularly those that address reducing the outdoor mosquito population.

Response:

We are citing references 9, 14 and 15 regarding ceiling netting and its potential reduce both indoor and peri domestic outdoor mosquito populations, with possible spillover effects on community-level mosquito abundance. We specifically, included the following edit (line 81): “, with possible spillover effects on community-level mosquito abundance [9, 14, 15].”

STUDY DESIGN AND PROCEDURES

Comment 5 (Line 105-106): "Briefly, we planned the study including three clusters of Llanchama households randomly selected." What is the basis for the randomization of clusters? Is it a comparison of house types (later referred to as types B, C, and D) within one cluster, or is it simply the number of houses grouped closely together at the study sites?

Response:

We revised the Methods as follows to explained cluster allocation (lines 217–218): “based on geographic proximity and housing stratification.”

Comment 6. There are no marks or clear boundaries for each cluster on the study map (see Supplementary Material S2).

Response:

Sadly, we could not obtain CC BY 4.0 permission for the original detailed maps, so we were forced to updated the map for location purposes only.

Comment 7. What is the distance between clusters?

Response:

We thank the reviewer for highlighting the need for clarification on spatial separation. Following his/her advise we have added the following clarification into the Methods section (lines 291–293): “The average distance between clusters was approximately 300–500 meters, sufficient to minimize contamination between groups given the short nightly flight range of Anopheles darlingi in this region (typically <1 km).

Comment 8. If the clusters did not have clear boundaries, it would be possible for houses from different clusters to be very close to each other. Considering that Anopheles can fly quite long distances, would there be any bias if Anopheles could not enter a treated house, but could fly to an untreated house next door?

Response:

We appreciate this thoughtful observation. While Anopheles darlingi is known to have a flight range of up to 1–2 kilometers under certain conditions, studies in the Peruvian Amazon indicate that its typical host-seeking range is much shorter (often <500 meters), and its movement is generally constrained by environmental features and host availability (PMID: 26223450). To minimize contamination and potential bias, households were grouped into clusters based on geographic proximity, but we ensured that within-cluster proximity was greater than between-cluster proximity. The average separation between clusters was 300–500 meters, and interventions were rolled out at the cluster level to preserve temporal and spatial separation inherent in the stepped-wedge design. While some edge effects are possible, our use of cluster-level analysis with random intercepts helped account for such potential inter-household influence. To address the reviewers comment we have added this consideration to the Discussion section (lines 509–515): “While the average separation between clusters was 300–500 meters, and households were grouped to minimize inter-cluster proximity, it is possible that mosquitoes deflected by treated homes migrated to untreated ones in neighboring clusters. Importantly, studies in the Peruvian Amazon indicate that its typical host-seeking range is much shorter (often <500 meters) [19], and its movement is generally constrained by environmental features and host availability. Furthermore, our use of cluster-level modeling with random intercepts helped mitigate this bias, though some residual inter-cluster influence may persist.”

Comment 9 (Line 111): Table 1 does not show how long it takes to install ceiling-mounted mosquito nets in households that received the intervention. Can installation be carried out in all households in a cluster simultaneously in a short time (less than one day)? Or did it take three months, including the installation process, for all households to ultimately benefit?

Response:

We revised the Methods (line 178-179) as follows to explain the installation timeline: “All houses in a cluster were remodeled within 3–5 days.”

STUDY INTERVENTION

Comment 10 (Lines 137–139): “Homes with large cracks or exposed sections in wooden plank walls had these areas covered with mosquito netting to reduce entry points further.” Are there any records of how many houses received additional improvements involving mosquito nets on the walls?

Response:

Thank you. We have edited the methods (line 170) as follows to explain the detail the documented additional wall netting: “(14 houses received this modification)”.

STUDY OUTCOMES

Comment 11 (Line 152–153): “Captured indoors and outdoors via human landing catches (HLCs).” This is not explained in detail:

• How many times were HLCs conducted on two consecutive nights in each cluster, for example, once a week or biweekly?

• How many sentinel houses were there, and where were they located within the cluster?

Response:

We thank the reviewer for pointing out the need to clarify the HLC sampling protocol. After reflecting about the reviewers’ insights, we decided to precise that we measured our entomological outcomes indoor and peridomestic, and not strictly outdoor. Consequently, we changed all the mentions to outdoor to peridomestic and specified that human landing catches (HLCs) were conducted biweekly at each household indoor and peridomestic throughout the baseline and intervention periods. We edited the study outcomes as follows (lines 195-197): “The study outcomes were measured during the first week of each study period (baseline, control 1, control 2, and control 3).”

STUDY PROCEDURES

Comment 12. Line 184–186: “Before the intervention was introduced, baseline entomological assessments were conducted in all households to determine the initial density of mosquitoes and the extent of human exposure to bites.” Did the HLCs conduct baseline entomological assessments in all 69 households?

Response:

We thank the reviewer for this important clarification. Yes, baseline entomological assessments were conducted in all 69 households via HLCs as specified in lines 224-225.

SAMPLE SIZE

Comment 13 (Line 210): What are the differences between house types A, B, C, and D? Could you please provide the classification criteria for house types A, B, C, and D, along with pictures if possible? You can include the explanation in Supplementary Section S2.

Response:

We added classification criteria for types A–D and photos in Supplementary Material S2 (caption and description updated, lines 149–156). Additionally, we followed the reviewer recommendation and included the explanation in the Supplementary section S2.

RESULTS

Comment 14. Lines 298–300: Outdoor BPN (Fig. 2D), indoor BPH (Fig. 2E), and outdoor *BPN (Fig. 2F). IQR should be BPH.

Response:

Corrections have been made in Figure 2 legend (line 345-346). Please notice that understanding the point raised about outdoor we corrected the manuscript to precise that the outdoor measures were peridomestic. We believe that reviewer concerns about misinterpreting outdoor with peridomestic were correct, so we addressed it by making these important clarifications across the manuscript.

Comment 15. Line 299: Interquartile Range; *CMMN: Ceiling-mounted Mosquito Nets (does not refer to Figure 2).

Response:

Corrections have been made in Figure 2.

Comment 16. Line 303–304: IQR (interquartile range); BPN (bites per person per night); *BPN (bites per person per hour); grey shadow (intervention period). (Should be 'BPH', bites per person per hour.)

Corrections have been made in Figure 2 legend (line 346).

DISCUSSION

Comment 17. The effect of CMMN on non-Anopheles mosquitoes is worth discussing, particularly given that the number of Anopheles mosquitoes decreased indoors and outdoors during the study period.

Response:

We added a paragraph on this (lines 454–462): “Furthermore, although our study primarily targeted Anopheles vectors, incidental reductions in other mosquito genera were observed during the study period, particularly indoors. Structural modifications such as ceiling netting likely disrupt entry pathways for both malaria and non-malaria vectors. This is supported by previous research demonstrating that improved housing design, screened ceilings, and closed eaves reduce indoor densities of Culex and Aedes mosquitoes, contributing to broader vector control benefits [21, 22]. Future studies should quantify the effect of CMMNs on non-Anopheles species to assess their potential for integrated vector management.”

III. Responses to Reviewer #2 Comments

ABSTRACT

Comment 1: Statistical analysis (how the data were analyzed) is missing in the abstract section. It is important to appear in the abstract section

Response:

We agree. Consequently, we revised the Abstract and edited accordingly (lines 32–34): “Analysis followed a stepped-wedge mixed-effects negative binomial model adjusting for clustering and time trends.”

Comment 2 (Line 40): I don’t think funding is part of the abstract. The authors should revise the journal submission guidelines if this adheres to these guidelines

Response:

Thank you. We deleted the funding mention.

INTRODUCTION SECTION

Comment 3 (Line 47-49 ): Authors have described the vector control tools such as ITNs and IRS. Could the authors describe the challenges that may be facilitating the persistence of

---

## [Decision Letter · Decision Letter 1]

9 Sep 2025

Efficacy of Ceiling-Mounted Mosquito Nets for Malaria Vector Control in a Peruvian Amazon Riverine Community: A Stepped-Wedge Cluster Randomized Trial

PONE-D-25-21398R1

Dear Dr. Quispe,

We’re pleased to inform you that your manuscript has been judged scientifically suitable for publication and will be formally accepted for publication once it meets all outstanding technical requirements.

Kind regards,

James Colborn

Academic Editor

PLOS ONE

Additional Editor Comments (optional):

Reviewer #1:

Reviewer #2:

Reviewers' comments:

Reviewer's Responses to Questions

**Comments to the Author**

1. If the authors have adequately addressed your comments raised in a previous round of review and you feel that this manuscript is now acceptable for publication, you may indicate that here to bypass the “Comments to the Author” section, enter your conflict of interest statement in the “Confidential to Editor” section, and submit your "Accept" recommendation.

Reviewer #1: All comments have been addressed

Reviewer #2: All comments have been addressed

2. Is the manuscript technically sound, and do the data support the conclusions?

Reviewer #1: Yes

Reviewer #2: Yes

3. Has the statistical analysis been performed appropriately and rigorously? 

Reviewer #1: I Don't Know

Reviewer #2: Yes

4. Have the authors made all data underlying the findings in their manuscript fully available?

Reviewer #1: Yes

Reviewer #2: Yes

5. Is the manuscript presented in an intelligible fashion and written in standard English?

Reviewer #1: Yes

Reviewer #2: Yes

6. Review Comments to the Author

Reviewer #1: I have reviewed the revised manuscript.

The authors have integrated their responses into the text.

I have no further comments.

Reviewer #2: the answers I provided above shows how this article is worthy to be published in this journal. Firstly, it indicates the new intervention that can be used to control mosquito-borne disease vectors especially malaria. Secondly, it indicates that new interventions can be combined with other core interventions such as ITNs and IRS to reduce contact between human and malaria vectors.

7. PLOS authors have the option to publish the peer review history of their article (what does this mean? ). If published, this will include your full peer review and any attached files.

**Do you want your identity to be public for this peer review?** For information about this choice, including consent withdrawal, please see our Privacy Policy .

Reviewer #1: No

Reviewer #2: No

---

## [Editor Report · Acceptance letter]

PONE-D-25-21398R1

PLOS ONE

Dear Dr. Quispe,

I'm pleased to inform you that your manuscript has been deemed suitable for publication in PLOS ONE. Congratulations! Your manuscript is now being handed over to our production team.

Kind regards,

on behalf of

Dr. James Colborn

Academic Editor

PLOS ONE